# The SMC5/6 Complex Represses the Replicative Program of High-Risk Human Papillomavirus Type 31

**DOI:** 10.3390/pathogens9100786

**Published:** 2020-09-25

**Authors:** Ryan T. Gibson, Elliot J. Androphy

**Affiliations:** 1Department of Microbiology and Immunology, School of Medicine, Indiana University, Indianapolis, IN 46202, USA; ryangibs@iu.edu; 2Department of Dermatology, School of Medicine, Indiana University, Indianapolis, IN 46202, USA

**Keywords:** papillomavirus, structural maintenance of chromosomes, SMC5/6, NSE3, E2, replication, transcription

## Abstract

The multi-subunit structural maintenance of chromosomes (SMC) 5/6 complex includes SMC6 and non-SMC element (NSE)3. SMC5/6 is essential for homologous recombination DNA repair and functions as an antiviral factor during hepatitis B (HBV) and herpes simplex-1 (HSV-1) viral infections. Intriguingly, SMC5/6 has been found to associate with high-risk human papillomavirus (HPV) E2 regulatory proteins, but the functions of this interaction and its role during HPV infection remain unclear. Here, we further characterize SMC5/6 interactions with HPV-31 E2 and its role in the HPV life cycle. Co-immunoprecipitation (co-IP) revealed that SMC6 interactions with HPV-31 E2 require the E2 transactivation domain, implying that SMC5/6 interacts with full-length E2. Using chromatin immunoprecipitation, we found that SMC6 is present on HPV-31 episomes at E2 binding sites. The depletion of SMC6 and NSE3 increased viral replication and transcription in keratinocytes maintaining episomal HPV-31, indicating that SMC5/6 restricts the viral replicative program. SMC6 interactions with E2 were reduced in the presence of HPV-31 E1, suggesting that SMC6 and E1 compete for E2 binding. Our findings demonstrate SMC5/6 functions as a repressor of the viral replicative program and this may involve inhibiting the initiation of viral replication.

## 1. Introduction

Human papillomaviruses (HPVs) are non-enveloped, circular double-stranded DNA viruses that infect basal keratinocytes of stratified squamous epithelia. Infection with certain “high-risk” HPV types (e.g., HPV-16 and -31) causes nearly all cervical cancers and an increasing number of oropharyngeal cancers [1]. While prophylactic vaccinations have reduced the incidence of HPV infection and attributable cancers, there is currently no cure for pre-existing HPV infection. As such, HPV remains a global health threat and a better understanding of HPV biology remains of significant medical importance for the identification of novel therapeutic targets.

The HPV genome is roughly eight kilobases in length, encoding for eight proteins essential for the viral life cycle that are classified as early (E1–E7) or late (L1 and L2), based upon their temporal expression during HPV infection. The HPV replicative program can be divided into three hypothetical phases: Establishment, maintenance, and amplification. During the initial establishment of infection, HPV E1 and E2 proteins, together with the host replicative machinery, replicate HPV genomes to low copy numbers in host cells. Following establishment, the HPV genome enters the maintenance phase, whereby viral genome copy numbers are maintained and replication occurs in synchrony with host cells. In the outer layers of the epithelium, the viral replicative program transitions from maintenance to amplification, whereupon viral genomes are replicated to thousands of copies per cell and encapsidated by L1 and L2 viral proteins within the nucleus. Throughout the HPV life cycle, the E2 protein is essential for regulating viral replication, transcription, and genome maintenance via interactions with both viral and host factors. For example, E2 binds to multiple sites flanking the HPV genome origin of replication to facilitate the recruitment of factors that either stimulate or repress viral genome replication [2,3].

The structural maintenance of chromosomes (SMC) family of multi-subunit protein complexes includes SMC1/3 (cohesin), SMC2/4 (condensin), and SMC5/6 [4]. The SMC5/6 complex is comprised of two SMC subunits (SMC5 and SMC6) and four non-SMC elements (NSE1-4), which join to form a ring-shaped structure used to topologically entrap DNA. Both SMC5 and SMC6 peptides fold back upon a central hinge region that connects to N- and C-terminal globular domains via a long coiled-coil arm. These SMC subunits form obligate heterodimers via the central hinge. Closure of the SMC5/6 complex into the ring-shape characteristic of all SMC protein complexes occurs by bridging SMC5 and SMC6 N- and C-termini by the kleisin subunit NSE4. NSE1 and NSE3 bind NSE4, forming the NSE1/3/4 sub-complex that enables the NSE3-dependent DNA binding activity of SMC5/6 [5]. Amongst SMC protein family members, SMC5/6 is unique in possessing intrinsic enzymatic activity via an NSE2 SP-RING SUMO ligase subunit that is bound to the coiled-coil arm of SMC5 [6]. SMC5/6 is required for multiple cellular functions, such as host DNA damage repair by homologous recombination [4], and is essential for cell viability [7]. Recently, antiviral roles have been described for SMC5/6, whereby SMC5/6 restricts transcription of hepatitis B [8,9,10,11] and herpes simplex-1 (HSV-1) viruses [11]. SMC5/6 has also been found to interact with both high- and low-risk HPV E2 proteins and is purportedly involved in HPV genome maintenance [12], but the significance of SMC5/6 interactions with E2 and its role during the HPV life cycle is not fully understood.

In this study, we examined interactions of the SMC5/6 complex subunits SMC6 and NSE3 with HPV-31 E2 and E1 proteins, as well as their roles during the maintenance phase of the HPV-31 replicative program. Previously reported antiviral functions of SMC5/6 during HBV and HSV-1 infections led us to hypothesize that SMC5/6 would function as a repressor of the HPV-31 life cycle. We found that the depletion of SMC6 and NSE3 resulted in increased HPV-31 replication and transcription, suggesting that the SMC5/6 complex represses the replicative program of HPV-31. Our data demonstrate that SMC6 is present on episomal HPV-31 genomes and associates with viral E2 proteins, but interactions with E2 are reduced in the presence of E1, indicating that SMC6 competes with E1 for E2 binding. This infers that SMC6 repression of the viral replicative program may involve inhibiting the initiation of viral replication by disrupting E2 interactions with E1. Taken together, our results point to a novel role for SMC5/6 as a repressor of the HPV-31 replicative program.

## 2. Results

The SMC5/6 complex (Figure 1A) subunit SMC6 reportedly co-precipitates with several HPV E2 proteins [12,13,14], so we began our investigation by attempting to confirm this interaction with HPV-31 E2. HEK293 cells were transfected with FLAG-SMC6, HPV-31 V5-E2, or both plasmids, and FLAG-SMC6 immunoprecipitation (IP) was then performed using the M2 anti-FLAG antibody. Both HPV-31 V5-E2 and the NSE3 positive control co-immunoprecipitated (co-IP) with SMC6 (Figure 1C). In the reciprocal experiment, the antibody to the V5 epitope pulled-down FLAG-SMC6, but not the SMC5/6 subunit NSE3 (Figure 1D). Structurally, HPV E2 is comprised of an N-terminal transactivation domain (TAD) connected to a C-terminal DNA binding/dimerization domain (DBD) by a poorly conserved linker region (Figure 1B). To identify the region of E2 required for these interactions, HEK293TT cells were transfected with FLAG-SMC6, FLAG-tagged full-length E2, or FLAG-tagged truncated E2 expressing the E2 hinge and DBD (Figure 1E). Co-IP was performed using the anti-SMC6 antibody and showed full-length HPV-31 FLAG-E2 and NSE3 co-IP with SMC6, but not truncated E2 (Figure 1F). These data demonstrate that SMC6, but not NSE3, associates with HPV-31 E2 and that these interactions require the E2 transactivation domain.

Previously, SMC5/6 has been shown to inhibit the transcription of HBV [8,9,10,11] and HSV-1 [11]. We questioned whether these antiviral functions of SMC6 and its interactions with HPV E2 indicated that SMC5/6 functions as a repressor of HPV-31 replication or transcription. We began investigating this using the siRNA knockdown of SMC6 and NSE3 in CIN612-9E cells (Figure 2A)—a cervical keratinocyte cell line derived from a patient CIN I lesion that maintains low copy numbers of episomal HPV-31 [15]. SMC6- and NSE3-depleted cells displayed an increased HPV-31 DNA content (Figure 2B) and viral E8^E2 transcript abundance (Figure 2C). SMC6 knockdown using an alternative siRNA also resulted in increased HPV-31 transcripts in CIN612-9E cells (Appendix A). We performed an analogous experiment in episomal maintaining NIKS/HPV-31 keratinocytes. Similar results were found, as the depletion of either SMC6 or NSE3 resulted in an increased HPV-31 DNA content in NIKS/HPV-31 (Figure 2D,E); however, only NSE3 knockdown increased E8^E2 viral transcripts (Figure 2F). Taken together, these data show that the transient depletion of SMC6 and NSE3 in CIN612-9E and NIKS/HPV-31 results in increased HPV-31 replication and transcription.

SMC6 has been shown to associate with nuclear episomal HBV cccDNA [8] and HSV-1 DNA [11], but how SMC5/6 localizes to viral DNA remains unclear. An SMC6 association with HPV genomes has not been reported. Similar to HBV cccDNA and HSV-1, HPV genomes are maintained as episomes adjacent to promyelocytic leukemia (PML) nuclear bodies [16], which are known sites of SMC5/6 localization [10]. The association of SMC6 with viral DNA and its interaction with HPV E2 led us to question whether SMC6 is present on episomal HPV-31 genomes. To answer this, we first validated our ability to immunoprecipitate SMC6 in episomal HPV-31 maintaining CIN612-9E cells to determine the suitability of the cell line for subsequent chromatin immunoprecipitation (ChIP) experiments. Total protein was isolated from CIN612-9E cells and SMC6 immunoprecipitation (IP) was performed, followed by immunoblotting using the anti-SMC6 antibody. SMC6 was detectable in input samples and IP samples where the anti-SMC6 antibody was used, but not negative control rabbit IgG (Figure 3A). SMC5/6 is present on DNA during the S phase of the cell cycle to facilitate DNA damage repair and restart stalled replication forks [17,18,19]. CIN612-9E cells were arrested in the S phase by a double-thymidine block to enrich for SMC6’s association with DNA, and ChIP was then performed using HPV-31 E2, SMC6, or rabbit IgG antibodies. HPV-31 E2 served as our positive control, as it is well-established that E2 binds episomal HPV genomes and our lab has previously demonstrated that E2 binds to ChIP at various sites along the long control region (LCR) of HPV-31 episomes [2,3]. ChIP and qPCR determined that HPV-31 E2 was at E2 binding sites within the HPV-31 LCR, but not the viral L1 region (Figure 3C). SMC6 was also not present in the HPV-31 L1 region, but was detected near E2 binding sites located towards the LCR 5′ region and near the viral ori and early promoter (Figure 3D). E2 is present on cellular DNA [20,21,22] and reportedly tethers HPV episomes to host common fragile sites [23]. We questioned whether SMC6 is associated with a cellular replication origin (GM-CSF) where E2 may be found. HPV-31 E2, but not SMC6, was present on the eukaryotic GM-CSF origin of replication (Figure 3B). The absence of SMC6 at the GM-CSF origin led us to utilize the GM-CSF locus as a negative control for SMC6 binding and to evaluate the enrichment of SMC6 on viral DNA. These data show that SMC6 is present on HPV-31 episomes at E2 binding sites located within the LCR in the 5′ region and near the viral ori and early promoter, but not the flanking L1 region.

HPV replication depends on E2-mediated recruitment of the viral E1 DNA helicase to the viral origin of replication, during which a transient E1/E2-ori ternary complex is formed prior to the release of E2 from the complex and viral genome [24,25,26,27]. The association of SMC6 with E2 and HPV-31 genomes, as well as its repression of viral replication, led us to speculate that SMC6 may interact with HPV-31 E1/E2 replicative complexes. To explore this, HEK293TT cells were transfected with FLAG-SMC6, HPV-31 V5-E2, and HPV-31 HA-E1 plasmids alone or in combination, and co-IP reactions were performed using the anti-SMC6 antibody. We observed HPV-31 FLAG-E2 and NSE3 co-IP with endogenous SMC6 (Figure 4A), but not HPV-31 HA-E1, when E1 was expressed alone or co-expressed with E2, or when FLAG-SMC6 was co-transfected with E1 and E2. The co-expression of E1 with FLAG-SMC6 and E2 reduced the amount of E2 pulled-down by SMC6. To confirm that SMC6 does not co-IP HPV-31 E1/E2 complexes or E1 alone, we performed an analogous experiment using the anti-HPV-31 E1 antibody for IP. The IP of HPV-31 HA-E1 pulled-down V5-E2, but not SMC6 or NSE3 (Figure 4B). The E1 co-IP of E2 was not influenced by the overexpression of FLAG-SMC6. Together, these data demonstrate that neither SMC6 nor NSE3 associate with HPV-31 E1 or E1/E2 complexes, but that E1 expression inhibits the SMC6 pulldown of E2.

HPV E2 proteins have been found to influence the host factor stability and transcriptional activity [28,29]. Having validated E2/SMC6 interactions led us to question whether E2 influenced the SMC6 abundance. To explore this, we generated HaCaT cells stably expressing HPV-31 FLAG-E2 or a control vector by G418 selection (Figure 5A). We observed no change in SMC6 protein levels in response to HPV-31 E2 expression. To confirm this, we examined the SMC6 protein expression in parental N/TERT cells and N/TERT/E2 cells stably expressing the HPV-16 E2 protein (Figure 5B). Similarly, no effect on SMC6 levels was found in N/TERT cells with HPV-16 E2 compared to control cells.

## 3. Discussion

In this study, we investigated SMC6 interactions with HPV proteins and explored the role of SMC5/6 during the maintenance phase of the HPV-31 life cycle. We observed SMC6 co-IP HPV-31 E2 and further identified that this interaction requires the E2 transactivation domain. While we cannot rule out a contribution of the E2 DNA-binding domain, these data infer that SMC6 interacts with full-length HPV-31 E2. Importantly, we found that the knockdown of SMC6 and NSE3 results in increased HPV-31 replication and transcription, implying that SMC5/6 represses these processes. We had speculated that the SMC5/6-mediated repression of HPV replication may occur through interactions of SMC6 with E1/E2 complexes as a mechanism utilized by E2 to prevent over-replication of viral genomes. This is unlikely as we did not co-immunoprecipitate SMC6 or NSE3 subunits with either HPV-31 E1/E2 complexes or E1. The association of SMC6 with E2 appeared to be reduced in the presence of E1 and together, these data imply that SMC6/E2 and E1/E2 complexes are exclusive. This suggests that SMC6 repression of the viral replicative program may involve inhibiting E2 recruitment of E1 to viral genomes for the initiation of replication. Our observations that viral replication increased in cells following the depletion of SMC6 and NSE3 support this explanation, as more E2 would be available to recruit E1 to viral genomes for initiating replication. Together, our study supports a role for the SMC5/6 complex as an endogenous antiviral factor and repressor of the HPV-31 replicative program.

SMC5/6 associates with host chromatin to mediate homologous recombination DNA repair, rescue stalled replication forks [30], and topologically entrap extrachromosomal DNA [31]. Additionally, SMC5/6 has been shown to associate with hepatitis B [8,11] and herpes simplex-1 [11] viral episomes. The purported localization of SMC6 to viral DNA and its association with E2 led us to speculate that SMC6 may be present on episomal HPV-31 genomes. Indeed, we found SMC6 on episomal HPV-31 genomes at E2 binding sites located towards the LCR 5′ region and proximal to the viral ori and early promoter, but not the L1 region or GM-CSF origin of replication. These data demonstrate that SMC6 localization on the viral genome is specific to the LCR, inferring that SMC5/6 repressor functions may occur at the viral genome. Mechanistically, the repression of HPV-31 replication and transcription could involve SMC5/6 influencing the accessibility of viral DNA through steric hindrance, the recruitment of histone modifying proteins, or interactions with E2 at the viral genome. It is also possible that E2 recruits SMC5/6 to viral episomes to regulate transcription and replication, as this is an established function of E2 [32].

The absence of NSE3 in the complex with E2 suggests that E2 may alter SMC5/6 complex formation by excluding NSE3. While this possibility accounts for the observed absence of NSE3, we believe it to be unlikely. SMC5/6 is comprised of a core SMC5/6 heterodimer and several non-SMC elements (NSE1-4). [4]. With the exception of NSE2, failure of any one subunit to join results in proteasomal degradation of the complex [6]. Should E2 interactions with SMC6 disrupt the association of NSE3 with the SMC5/6 complex, NSE3 would be degraded. We did not find the NSE3 abundance to be altered in any experiment where cells were co-transfected with E2. This implies that the SMC5/6 complex was intact and NSE3 was not degraded. Furthermore, the amount of NSE3 pulled down by SMC6 co-IP was unchanged when E2 was present, inferring that SMC6 interactions with NSE3 were not disrupted by E2. However, without internal loading controls, we cannot exclude the possibility that low levels of NSE3 were degraded and E2 sequesters a fraction of SMC6 to form a separate complex that does not include NSE3.

SMC5/6 has previously been reported to inhibit the life cycle of hepatitis B (HBV) by repressing transcription of the non-replicating, covalently closed circular DNA (cccHBV) intermediate form of HBV [8,9,10,11]. Our data support an antiviral function for SMC5/6 that extends to repression of the HPV-31 replicative program, as SMC6 and NSE3 depletion enhanced viral replication and transcription. However, how SMC5/6 represses the replicative program of HPV-31 is less clear. Unlike cccHBV, HPV episomes are actively transcribed and replicated throughout the viral life cycle. We found that both processes were increased in response to SMC6 and NSE3 loss, suggesting that SMC5/6 represses one, or both, processes. For example, it is possible that SMC5/6 functions as a repressor of HPV transcription and that the observed increase in HPV replication was due to increased transcription. Conversely, SMC5/6 may repress HPV replication, but not transcription. If SMC5/6 functions as a repressor of HPV replication alone, then the knockdown of SMC6 and NSE3 would result in increased replication and thus more viral DNA would be available to be transcribed. Consequently, the observed increase in the viral transcript abundance would not be resultant from an increase in the rate of viral gene transcription, but due to more viral genomes being transcribed. Future studies are needed to discern if the repressive functions of SMC5 and NSE3 during the life cycle of HPV are specific to replication, transcription, or both.

While our findings are in agreement with others describing an antiviral function for SMC5/6 [8,9,10,11], they contrast with a previous report in which SMC5/6 was not found to repress HPV-31 replication or transcription, but was instead necessary for the maintenance of viral genomes [12]. This discrepancy has several possible explanations. For example, our experiments relied upon the transient depletion of SMC5/6 subunits SMC6 and NSE3, as the sustained depletion of SMC5/6 subunits has been found to result in spontaneous DNA damage and chromosomal abnormalities and to be lethal in both non-cancerous and cancerous cell types [7]. It is conceivable that all of these factors contributed to the reported loss of HPV episomes over time in cells with a near complete reduction of SMC6. However, we cannot dismiss the possibility that SMC5/6 may also be involved in the maintenance of genomes.

HPV E2 proteins have been reported to influence the gene expression and protein stability of host factors [28,29]. Neither HPV-31 nor HPV-16 E2 proteins altered SMC6 levels in human keratinocytes. While this finding agrees with another reporting that HPV-5/-18 and BPV-1 E2 proteins do not influence the SMC6 abundance [12], it contrasts with studies of hepatitis B virus (HBV) wherein the master regulatory protein of HBV—HBx—was found to target SMC5/6 for degradation shortly after infection [8,9]. It is possible that this is due to differences in the cell type infected and differing roles of SMC5/6 in the life cycles of these viruses. The replicative program of HPV is dependent upon maintaining the actively dividing state of infected basal keratinocytes and therefore, while the loss of SMC5/6 may be tolerable in non-dividing HBV-infected hepatocytes, actively dividing HPV-infected keratinocytes would require a functional SMC5/6 complex for sustaining cell viability.

In conclusion, our study has identified a novel role for the SMC5/6 subunits SMC6 and NSE3 in the life cycle of HPV-31. Our observation that transient SMC6 and NSE3 depletion results in increased HPV-31 replication and transcription implies that SMC5/6 represses the replicative program of HPV-31. The function of SMC5/6 in hepatitis B virus and herpes simplex-1 viral infections appears to be the repression of viral DNA transcription; however, we have not determined whether the antiviral functions of SMC6 and NSE3 during HPV infection are specific to transcription, replication, or both. While we have demonstrated that SMC6 is present on episomal HPV-31 genomes and interacts with HPV E2 proteins, the mechanism by which SMC5/6 represses the viral replicative program remains unclear. SMC5/6-mediated repression may involve inhibiting the E2 recruitment of E1 or other replication factors to the viral ori.

## 4. Materials and Methods

*Plasmids, siRNA, and Antibodies.* The plasmids used were FLAG-SMC6/pcDNA3 (GenScript, Piscataway, NJ, USA), HPV-31 FLAG-E2/pcDNA3, HPV-31 FLAG-E2^mt^ (residues 205-372; [33]), HPV-31 V5-E2/pcDNA3, HPV-31 HA-E1/pcDNA3, and pcDNA3. The siRNA depletion of SMC5/6 subunits used siScramble DsiRNA (IDT; 51-01-19-08), siSMC6 (IDT; hs.Ri.SMC6.13.2), and siNSE3 (IDT; hs.Ri.NDNL2.13.1). Appendix A SMC6 siRNA knockdown used esiGFP (Sigma-Aldrich, St. Louis, MO, USA; EHUEGFP) and esiSMC6 (Sigma-Aldrich, St. Louis, MO, USA; EHU026931). Antibodies included mouse M2 anti-FLAG (Sigma-Aldrich, St. Louis, MO, USA), mouse anti-V5 (Invitrogen, Waltham, MA, USA), rabbit anti-V5 (Cell Signal Technologies, Danvers, MA, USA), rabbit anti-NSE3 (Abcam, Cambridge, UK), mouse anti-SMC6 (Abgent, San Diego, CA, USA), rabbit anti-SMC6 (Bethyl, Montgomery, TX, USA), rat anti-HPV-31 E1 ([2]), mouse DM1a anti-α-tubulin (Sigma-Aldrich, St. Louis, MO, USA), mouse anti-β-actin (Sigma-Aldrich, St. Louis, MO, USA), mouse TVG-261 anti-HPV-16 E2 (Abcam, Cambridge, UK), sheep anti-HPV-16 E2 ([34]), and rabbit IgG (Jackson, Lansing, MI, USA). Chromatin immunoprecipitation (ChIP) experiments employed rabbit anti-HPV-31 E2 ([2]; 10008), rabbit anti-SMC6 (Bethyl, Montgomery, TX, USA), rat anti-HPV-31 E1, and rabbit IgG (Jackson, Lansing, MI, USA) antibodies.

*Cell Culture.* HEK293 and HEK293TT cells were maintained in Dulbecco’s Modified Eagle Medium (DMEM) (Life-Technologies, Carlsbad, CA, USA) supplemented with 10% fetal bovine serum (FBS) (Atlas Biologicals, Fort Collins, CO, USA) and 1X penicillin/streptomycin (Life-Technologies, Carlsbad, CA, USA) at 37 °C and 5% CO_2_. HaCaT/HPV-31 FLAG-E2 and HaCaT/pcDNA3 were cultured in DMEM/10% FBS and 200 μg/mL G418 (Life-Technologies, Carlsbad, CA, USA). CIN612-9E cells (maintaining episomal HPV-31b) were cultured in E-media (3:1 [v/v] DMEM/Ham’s F-12/5% FBS/0.4 μg/mL hydrocortisone/0.1 nM cholera toxin/5 μg/mL transferrin/5 ng/mL hEGF/5 μg/mL insulin/adenine μg/mL/1X penicillin/streptomycin) with mitomycin C-treated J2 3T3 fibroblast feeders (J2s). NIKS/HPV-31 (episomal HPV-31) were cultured in F-media (3:1 [v/v] Ham’s F-12/DMEM/5% FBS/0.4 μg/mL hydrocortisone/0.1 nM cholera toxin/5 ng/mL hEGF/5 μg/mL insulin/24.3 μg/mL adenine/1X penicillin/streptomycin) with 250 μg/mL G418 and mitomycin C-treated J2s. N/TERT and N/TERT/HPV-16 E2 cells were cultured in KSFM supplemented with bovine pituitary extract, hEGF, and 4 μg/mL hygromycin B (Sigma-Aldrich, St. Louis, MO, USA).

*Co-IP and immunoblotting.* HEK293 or HEK293TT cells were transfected using polyethylenimine (PEI; 2 mg/mL, Sigma-Aldrich, St. Louis, MO, USA) and then lysed 48 h later in low-salt NP-40 lysis buffer (0.5% NP-40/150 mM NaCl/50 mM Tris-HCl (pH 8)/10% glycerol) and protease inhibitor cocktail (Sigma-Aldrich, St. Louis, MO, USA). Lysates were rotated for 1 h at 4 °C with benzonase (Millipore) and spun and the supernatant was then collected. For FLAG co-IP experiments, samples were rotated overnight at 4 °C with M2 agarose bead slurry (Sigma-Aldrich, St. Louis, MO, USA). SMC6 co-IPs were performed as in V5 and E1 co-IPs, but using either rabbit anti-SMC6 (Bethyl, Montgomery, TX, USA) or rabbit IgG control (Jackson, Lansing, MI, USA) antibodies. Beads were washed four times, alternating high-salt (0.5 M) and low-salt (150 mM) NP-40 lysis buffer washes, and boiled, and SDS-PAGE with the semi-dry transfer of proteins to PVDF membranes was then performed. Membranes were blocked in 5% milk/1X PBS/Tween-20 (0.1%) and then probed overnight at 4 °C with primary antibodies. Protein detection was performed using HRP-conjugated 2° antibodies and chemiluminescence substrates (GE). Total protein was isolated by lysis in 10% SDS/150 mM NaCl lysis buffer, samples were boiled, and Western blot analysis was performed as described above.

*Chromatin immunoprecipitation (ChIP).* ChIP was performed using the ChIP-IT Express Enzymatic kit (Active Motif, Carlsbad, CA, USA), according to the manufacturer’s protocols. qPCR was performed using SsoFast Evagreen Mastermix (Bio-Rad, Hercules, CA, USA) with primers listed in Table 1. Data were analyzed using BioRad CFX with C_t_ values normalized to input samples and fold change calculated relative to control IgG samples (set to 1). Statistical significance was determined by a comparison of fold change values against negative DNA binding sequences or control IgG using a one-tailed Student’s *t*-test.

*qPCR DNA replication assay.* The transient depletion of SMC5/6 subunits was performed using CIN612-9E and NIKS/HPV-31 cells reverse-transfected with 15 or 20 nM siRNA, respectively, using Lipofectamine 2000 (Life-Technologies, Carlsbad, CA, USA), as per the manufacturer’s protocol. The following day, media were changed and cells returned to incubation for an additional 24 h. DNA was then isolated by phenol/chloroform extraction and qPCR performed as described above with C_t_ values normalized to β-actin and fold change calculated relative to control groups. All experiments were performed in triplicate and statistical significance was determined by a two-tailed Student’s *t*-test.

*RT-qPCR.* RNA was isolated using PureLink RNA Micro kits (Invitrogen, Waltham, MA, USA). RQ1 DNase (Promega, Madison, WI, USA) was used to remove genomic DNA contaminants. Reverse transcription was performed using SuperScript III reverse transcriptase (Invitrogen, Waltham, MA, USA), qPCR was performed as described above with C_t_ values normalized to GAPDH or β-actin, and fold change was calculated relative to control groups and primers listed in Table 1. All experiments were performed in triplicate and statistical significance was determined by a two-tailed Student’s *t*-test.

**Table 1 pathogens-09-00786-t001:** HPV-31 ChIP and qPCR primers used in Figure 2 and Figure 3 and Appendix A.

Primer Set	Sequence (5′>3′)	Reference
**HPV-31 L1**	F: cacctccctcaggttctttg R: atggatcttccttgggcttt	[2]
**HPV-31 ESBS (5′)**	F: cctgctcctcccaatagtca R: ggaccgggtgtacaactttt	[2]
**HPV-31 E2BS (3′)**	F: ttgtgcaaacctacagacgcca R: agcttagttcatgcaatttccgagg	n/a
**HPV-31 gDNA**	F: gttctgcggtttttggtttc R: tgttggcaaggtgtgttagg	[2]
**GM-CSF**	F: acccagtccacctcactaat R: ttggtttcctttcagagcct	[35]
**HPV-31 E2**	F: agcgttgtcagtatcaaaggc R: gctgcattgtccagtcctcat	n/a
**HPV-31 E8^E2**	F: gtggaaacgcagcagatggta R: ttcgatgtggtggtgttgttg	n/a
**HPV-31 E1**	F: agaggctgtgcaggttctaa R: catctgctgcgtttccactt	n/a
**HPV-31 E1^E4**	F: tgttaatgggctcatttggaa R: ggttttggaattcgatgtgg	[36]
**SMC6**	F: gttggcgaaatgaaccggag R: cttgtcttggccttttggca	n/a
**β-actin**	F: gaggcactcttccagccttc R: cggatgtccacgtcacactt	[2]
**GAPDH**	F: cgcaggccggatgtgttc R: acgaccaaatccgttgactcc	n/a

## Figures and Tables

**Figure 1 pathogens-09-00786-f001:**
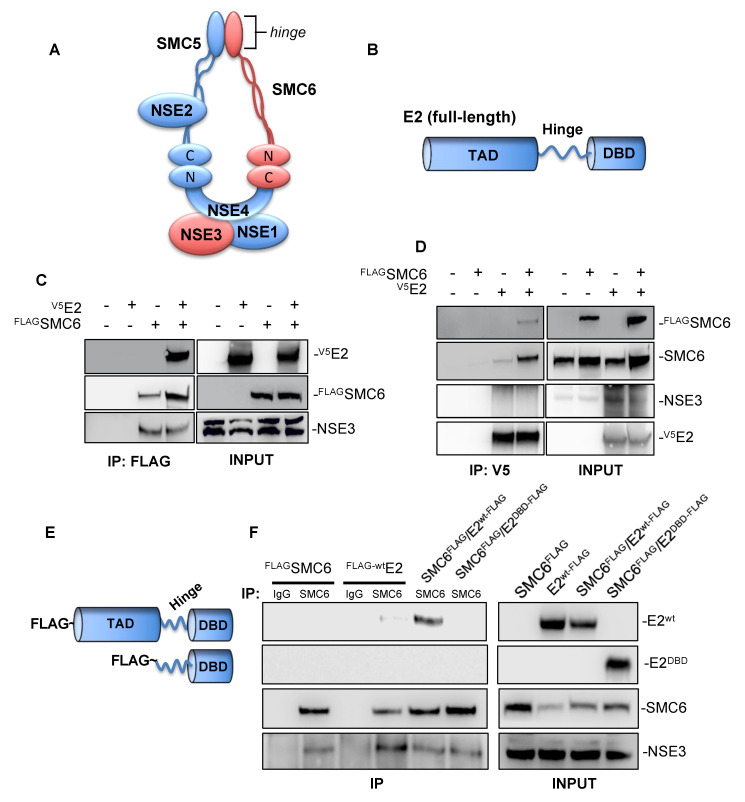
Structural maintenance of chromosomes (SMC) 6 co-immunoprecipitates human papillomavirus (HPV)-31 E2 in HEK293/TT cells. (**A**) Schematic representation of the eukaryotic SMC5/6 complex. (**B**) Diagram depicting the protein structure of full-length PV E2. The E2 N-terminal transactivation domain (TAD) is connected by a central hinge to a C-terminal DNA-binding domain (DBD). (**C**) FLAG-tagged SMC6 and HPV-31 V5-E2 were transfected into HEK293 cells. Forty-eight hours later, immunoprecipitation was performed using mouse M2 anti-FLAG agarose beads, and immunoblotting was then performed using mouse M2 anti-FLAG, mouse anti-V5, and rabbit anti-NSE3 (positive control) antibodies. (**D**) FLAG-tagged SMC6 and HPV-31 V5-E2 were transfected into HEK293TT cells. Forty-eight hours later, immunoprecipitation was performed using rabbit anti-V5 antibodies bound protein A/G sepharose beads, and immunoblotting was then performed using mouse M2 anti-FLAG, rabbit anti-SMC6, rabbit anti-V5, and rabbit anti-NSE3 antibodies. (**E**) Schematic of FLAG-tagged HR-HPV-31 full-length E2^wt^ (top) and truncated HPV-31 E2^DBD^ (bottom). (**F**) FLAG-SMC6, full-length FLAG-tagged HPV-31 E2, and FLAG-tagged HPV-31 truncated E2 were transfected into HEK293TT cells. Forty-eight hours later, immunoprecipitation was performed using either rabbit IgG or rabbit anti-SMC6, and immunoblotting was then performed using mouse M2 anti-FLAG, rabbit anti-SMC6, and rabbit anti-NSE3 antibodies.

**Figure 2 pathogens-09-00786-f002:**
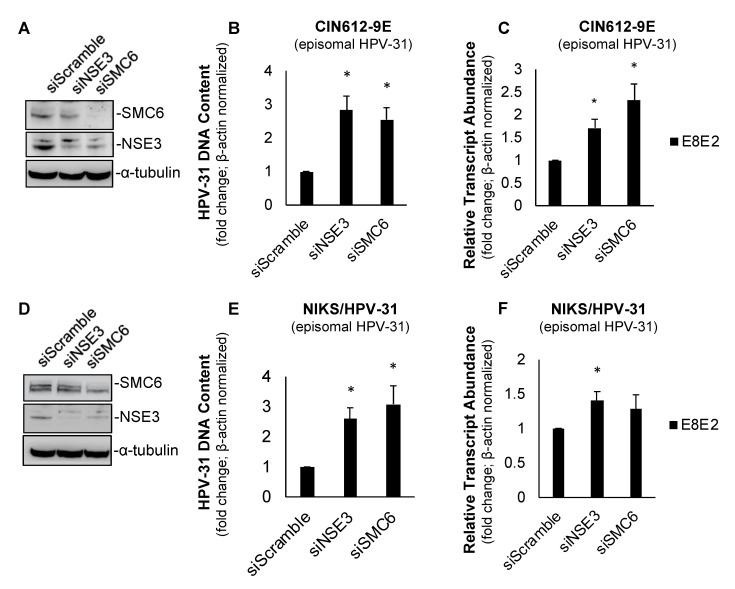
SMC6 and non-SMC (NSE)3 repress HPV replication and transcription in CIN612-9E and NIKS/HPV-31 cells. (**A**) Forty-eight hour siRNA knockdown of SMC6 and NSE3 resulted in a decreased target protein abundance in CIN612-9E cells. (**B**) The transient depletion of SMC6 increased the HPV-31 DNA content in CIN612-9E cells following 48 h siRNA knockdown. C_t_ values were normalized to β-actin and fold change is shown relative to the control (set to 1; *n* = 3). (**C**) As in (**B**), with RNA transcript abundance C_t_ values normalized to β-actin and fold change relative to the control (set to 1; *n* = 3). (**D**) As in (**A**), with NIKS/HPV-31 cells. (**E**) As in (**B**), with NIKS/HPV-31 cells. (**F**) As in (**C**), with NIKS/HPV-31 cells. All experiments were performed in triplicate and statistical significance was determined by a two-tailed Student’s *t*-test, where * = *p* ≤ 0.05.

**Figure 3 pathogens-09-00786-f003:**
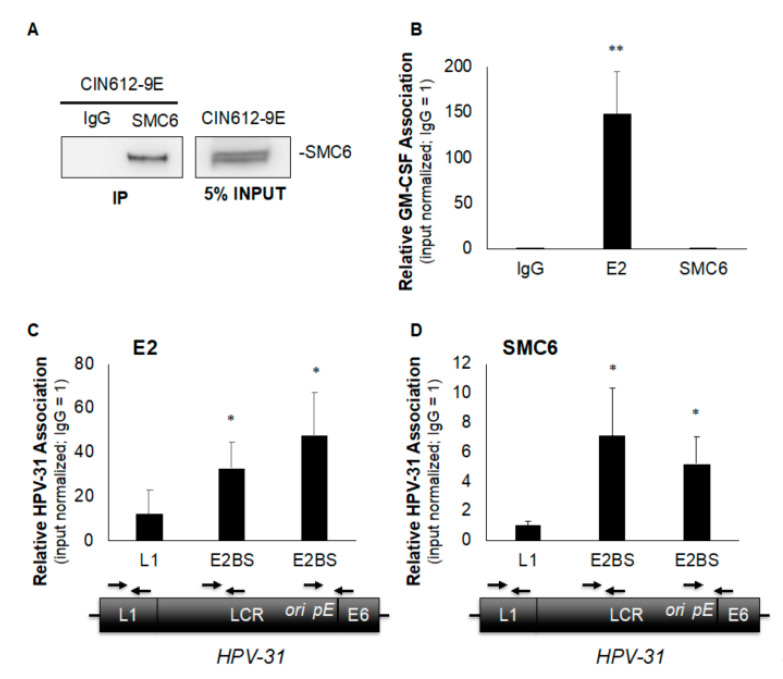
SMC6 associates with episomal HPV-31 genomes in CIN612-9E cells. (**A**) Total protein was isolated from CIN612-9E cells and immunoprecipitation was performed overnight using either rabbit anti-SMC6 or control rabbit IgG antibodies, and immunoblotting was then conducted using the rabbit anti-SMC6 antibody. (**B**) CIN612-9E cells were synchronized in the S phase by a double-thymidine block and chromatin immunoprecipitation (ChIP) was performed using rabbit anti-HPV-31 E2, rabbit anti-SMC6, or control rabbit IgG antibodies. qPCR was conducted using primers to the cellular replication origin (GM-CSF) of replication with primers listed in Table 1. (**C**) As in (**B**), but examining E2’s presence in the viral L1 region, at the 5′ LCR E2 binding site, and at the E2 binding site proximal to the ori and early promoter (pE) of the HPV-31 genome. Approximate locations on the viral genome and corresponding primers are shown below their respective sites on the horizontal axis. (**D**) As in (**C**), with SMC6. C_t_ values were normalized to the input with control IgG set to 1 and values are expressed as the mean fold change over IgG +/− SEM. ChIP was performed at least four independent times for all locations examined and statistical significance was determined by a one-tailed Student’s *t*-test compared to control IgG, with * *p* ≤ 0.05 and ** *p* ≤ 0.005.

**Figure 4 pathogens-09-00786-f004:**
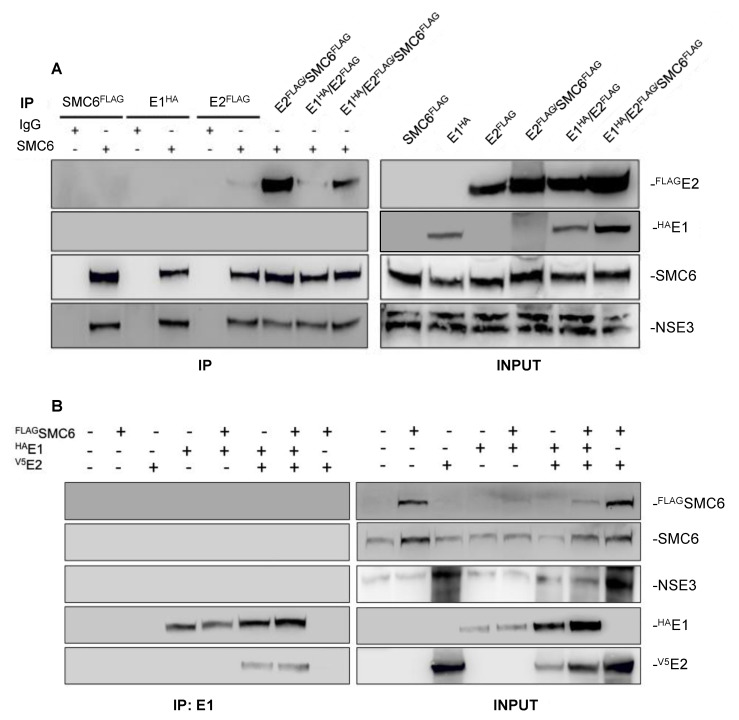
SMC6 and NSE3 do not co-IP HPV-31 E1 or E2/E1 complexes. (**A**) FLAG-SMC6, HPV-31 FLAG-E2, and HPV-31 HA-E1 were transfected into HEK293TT cells. Forty-eight hours later, immunoprecipitation was performed using either rabbit IgG or rabbit anti-SMC6, and immunoblotting was then performed using anti-SMC6, rabbit anti-NSE3, mouse M2 anti-FLAG, or rat anti-HPV-31 E1 antibodies. (**B**) FLAG-SMC6, HPV-31 V5-E2, and HPV-31 HA-E1 were transfected into HEK293TT cells. Forty-eight hours later, immunoprecipitation was performed using rat anti-HPV-31 E1, and immunoblotting was then performed using anti-SMC6, rabbit anti-NSE3, mouse M2 anti-FLAG, rabbit anti-V5, or rat anti-HPV-31 E1 antibodies. All experiments were performed three independent times.

**Figure 5 pathogens-09-00786-f005:**
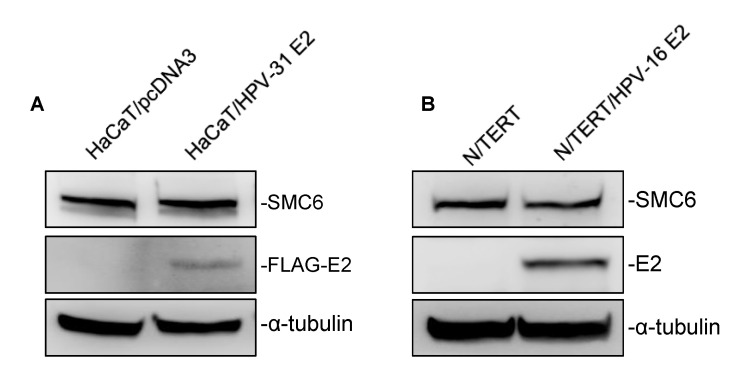
HPV-31/-16 E2 do not affect the SMC6 stability. (**A**) Total protein was isolated from the control and HPV-31 FLAG-E2 expressing HaCaT cells, and immunoblotting was then performed using mouse M2 anti-FLAG, rabbit anti-SMC6, and mouse DM1A anti-α-tubulin antibodies. (**B**) As in (**A**), but in parental N/TERT and N/TERT cells stably expressing HPV-16 E2 with immunoblotting using the mouse TVG-261 anti-HPV-16 E2 antibody. All experiments were performed in triplicate.

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
