# Peer review of "The SMC5/6 Complex Represses the Replicative Program of High-Risk Human Papillomavirus Type 31"

_pathogens, 2020, doi:10.3390/pathogens9100786_

Round 1
Reviewer 1 Report
The study investigates the observation that a variety of papillomavirus E2 proteins interact with the SMC5/6 proteins. A previous study implied SMC5/6 in the long-term maintenance of HPV31 genomes in tissue culture. This study finds that a knock-down of SMC6 increases HPV31 replication and the levels of the E8^E2 transcript. Furthermore, SMC6 interacts with viral genomes. Interestingly, in the presence of HPV31 E1 the interaction of E2 with SMC6 is reduced which points to a competition between SMC6 and E1. In summary this a well-executed study that improves our understanding of the consequences of the E2-SMC6 interaction.
Author Response
Reviewer 1 Comment: The study investigates the observation that a variety of papillomavirus E2 proteins interact with the SMC5/6 proteins. A previous study implied SMC5/6 in the long-term maintenance of HPV31 genomes in tissue culture. This study finds that a knock- down of SMC6 increases HPV31 replication and the levels of the E8^E2 transcript. Furthermore, SMC6 interacts with viral genomes. Interestingly, in the presence of HPV31 E1 the interaction of E2 with SMC6 is reduced which points to a competition between SMC6 and E1. In summary this a well- executed study that improves our understanding of the consequences of the E2-SMC6 interaction.
Authors' Response: We thank Reviewer 1 for taking their time to evaluate our manuscript. Reviewer 1 had no additional points to be addressed.
Reviewer 2 Report
This manuscript examined the interaction between SMC5/6 complex and E2 and proposed a novel pathway regulating papillomavirus type 31 replication by interrupt E1/E2 interaction.
Major:
- Figure 2.
- Because siRNA is notorious for its off-target effect, it would be great to show data from another unique siRNA targeting SMC6 or an rescue assay, especially when knockdown data contradicts the previous report.
- (L126-128): “All experiments were performed in triplicate and statistical significance determined by two-tailed (CIN612-9E) or one-tailed (NIKS/HPV-31) Student’s T-test...”
- CIN612-9E was analyzed with a two-tailed T-test, while NIKS/HPV-31 was analyzed with a one-tailed T-test. Because the mean is not dramatically different in the latter experiment (Fig. 2F), it may be improper to test NIKS/HPV with a one-tailed T-test.
- T-test requires two populations with the same variance, and siScramble group has an SD of 0; the student T-test may not be the most suitable test for this experiment. Equal variance should be tested, or the statistical significance should be determined by a test without variance requirement.
- L217-218: “Our data showed that the association of SMC6 with E2 was reduced in the presence of E1. This suggests that SMC6 repression of the viral replicative program may involve inhibiting initiation of viral replication by disrupting E2 interactions with E1. “
- Figure 4B shows that SMC6 does not disrupt E2 interaction with E1.
Minor:
- L175-177: “We observed HPV-31 V5-E2 and NSE3 co-IP with SMC6 (Figure 4a), but not HPV-31 HA-E1 when E1 was expressed alone, co-expressed with E2, or when FLAG-SMC6 was co-transfected with E1 and E2. “
- Fig. 4A are labeled as FLAG-E2.
Author Response
Reviewer 2 Comment: Because siRNA is notorious for its off-target effect, it would be great to show data from another unique siRNA targeting SMC6 or a rescue assay, especially when knockdown data contradicts the previous report.
Authors' Response: We used a second siRNA (Sigma-Aldrich; esiSMC6, EHU026931) to knockdown SMC6 that produced similar results in that transcript abundance of HPV31 E2, E8E2, E1 and E1E4 were increased following 48hr knockdown of SMC6 as compared to control (Sigma-Aldrich; esiGFP, EHUEGFP). Additionally, knockdown of NSE3 (SMC5/6 subunit required for DNA binding) using siNSE3 produced similar results to those found with SMC6 knockdown. This is now stated in the text.
Reviewer 2 Comment: (L126-128): “All experiments were performed in triplicate and statistical significance determined by two-tailed (CIN612-9E) or one-tailed (NIKS/HPV-31) Student’s T-test...”
CIN612-9E was analyzed with a two-tailed T-test, while NIKS/HPV-31 was analyzed with a one-tailed T-test. Because the mean is not dramatically different in the latter experiment (Fig. 2F), it may be improper to test NIKS/HPV with a one-tailed T-test.
T-test requires two populations with the same variance, and siScramble group has an SD of 0; the student T-test may not be the most suitable test for this experiment. Equal variance should be tested, or the statistical significance should be determined by a test without variance requirement.
Authors' Response: Statistical significance for NIKS/HPV-31 was determined using twotailed Student’s T-test instead of one-tailed Student’s T-test. NSE3 knockdown remained statistically significant (p = 0.042) and SMC6 was not statistically significant (p = 0.248) in Figure 4f. Two-tailed Student’s T-test showed that knockdown of NSE3 (p =0.0122) and SMC6 (p = 0.026) resulted in a statistically significant increase in viral DNA content in Figure 4e. Our initial use of one-tailed Student’s T-test was selected due to previous data showing that knockdown of SMC6 and NSE3 increased viral replication and transcription in human keratinocytes. Data and text now state CIN612-9E and NIKS/HPV-31 knockdown
results as analyzed by two-tailed Student’s T-test.
Reviewer 2 Comment: L217-218: “Our data showed that the association of SMC6 with E2 was reduced in the presence of E1. This suggests that SMC6 repression of the viral replicative program may involve inhibiting initiation of viral replication by disrupting E2 interactions with E1. “
Figure 4B shows that SMC6 does not disrupt E2 interaction with E1.
Authors' Response: Figure 4b (input lane 7) shows increased HPV-31 V5-E2 expression compared to lane 6 (input). Despite the increased abundance of V5-E2 in lane 7 (in which SMC6 is overexpressed) there was no corresponding increase in the amount of E2 pulled down by E1 in IP lane 7 as compared to lane 6. This suggests that E1 and SMC6 compete for binding of E2.
Reviewer 2 Comment: L175-177: “We observed HPV-31 V5-E2 and NSE3 co-IP with SMC6 (Figure 4a), but not HPV-31 HA-E1 when E1 was expressed alone, co-expressed with E2, or when FLAG- SMC6 was co-transfected with E1 and E2. “
Fig. 4A are labeled as FLAG-E2.
Authors' Response: Sentence has been changed to “We observed HPV-31 FLAG-E2 and NSE3 co-IP with endogenous SMC6 (Figure 4a), but not with HPV-31 HA-E1 when E1 was expressed alone, co-expressed with E2, or when FLAG-SMC6 was cotransfected with E1 and E2.” Figure 4a has been corrected to “FLAG-SMC6, HPV-31 FLAG-E2 and HPV-31 HA-E1 were transfected into HEK293TT cells.”
Round 2
Reviewer 2 Report
“Response 3: Figure 4b (input lane 7) shows increased HPV-31 V5-E2 expression compared to lane 6 (input). Despite the increased abundance of V5-E2 in lane 7 (in which SMC6 is overexpressed), there was no corresponding increase in the amount of E2 pulled down by E1 in IP lane 7 as compared to lane 6. This suggests that E1 and SMC6 compete for binding of E2.”
- It might be odd to interpret IP results in this way. Two variables are changing in this scenario. (E2 expression level; SMC6)
- It is convincing that E1 and SMC6 compete for binding of E2. However, when the affinity of E1-E2 is significantly higher than E2-SMC6, SMC may not be able to disrupt E1-E2 interaction.
- It would be great to show in Fig. 4b that overexpress SMC6 can disrupt E1/E2. Including another Figure 4b independent repeat in which Input Lane 6 and 7 are similar would strengthen the model.
Author Response
Reviewer 2 Comment: “Response 3: Figure 4b (input lane 7) shows increased HPV- 31 V5-E2 expression compared to lane 6 (input). Despite the increased abundance of V5-E2 in lane 7 (in which SMC6 is overexpressed), there was no corresponding increase in the amount of E2 pulled down by E1 in IP lane 7 as compared to lane 6. This suggests that E1 and SMC6 compete for binding of E2.”
It might be odd to interpret IP results in this way. Two variables are changing in this scenario. (E2 expression level; SMC6)
It is convincing that E1 and SMC6 compete for binding of E2. However, when the affinity of E1-E2 is significantly higher than E2-SMC6, SMC may not be able to disrupt E1-E2 interaction.
It would be great to show in Fig. 4b that overexpress SMC6 can disrupt E1/E2. Including another Figure 4b independent repeat in which Input Lane 6 and 7 are similar would strengthen the model.
Authors' Response: We agree with Reviewer 2 and have revised the manuscript to no longer state that SMC6 disrupts E2-E1 interactions as modified in Lines 237-244 and Lines 320-321 (highlighted yellow).